# Mitigating Generative Privacy Risks of Diffusion Models via Mixed Self-Synthesized Data Fine-tuning

## Abstract

Diffusion models (DMs) have demonstrated exceptional performance across various generative tasks, yet they also face significant security and privacy concerns, such as Membership Inference Attacks (MIAs), where adversaries attempt to determine whether specific images were part of the DM's training set. These threats present serious risks, particularly as pre-trained DMs are increasingly accessible online. To address these privacy concerns, we begin by investigating how fine-tuning DMs on a manipulated self-synthesized dataset affects their generative privacy risks, and have the following observations: (1) DMs fine-tuned solely on self-synthesized clean images are more vulnerable to privacy attacks (2) DMs fine-tuned on perturbed self-synthesized images become more robust against privacy attacks but exhibit degraded image generation quality. Based on the observations, we propose MixSyn, a simple and effective framework designed to mitigate privacy risks by fine-tuning DMs on a mixed self-synthesized dataset, which is a mixture of clean and perturbed synthetic images. Extensive experimental results demonstrate that our method significantly mitigates the generative privacy risks of DMs while preserving their original image generation quality.

## 1 Introduction

Text-to-Image Diffusion Models (DMs), such as Stable Diffusion (Rombach et al., 2022), DALL-E 3 (Ramesh et al., 2022), and Imagen (Saharia et al., 2022), have demonstrated outstanding performance in generating high-quality images. With the rapid advancements in DMs, their usage has significantly expanded across individuals and organizations for various applications. For instance, Stable Diffusion v1.4 has been downloaded more than 8 million times from the Huggingface repository, while Midjourney currently serves over a million users (Fatunde & Tse, 2022).

While diffusion models (DMs) demonstrate impressive capabilities in generating high-quality images, their deployment raises serious privacy concerns and copyright issues. Recent studies (Duan et al., 2023; Kong et al., 2023; Abascal et al., 2023) have identified that DMs are vulnerable to privacy leakage, particularly through Membership Inference Attacks (MIAs) (Shokri et al., 2016). MIAs attempt to determine whether a given data sample was part of the model's training set (member sample) or came from the hold-out set (non-member sample). When MIAs target DMs, they can potentially expose sensitive or private images used during pretraining, such as personal profile photos, medical images, or proprietary data from commercial entities. As artists increasingly form unions to combat the unauthorized use of their works by commercial generative models, the need for thorough security assessments and risk evaluations prior to the public release of these models becomes ever more pressing. Unfortunately, to make things worse, recent work S2L (Li et al., 2024) reveal that fine-tuning DMs on clean self-synthesized images could further amplify the power of MIA and make DMs more vulnerable to privacy attacks. Their findings show that the privacy risk with diffusion models is even more severe than previously recognized, and it is even more challenging to develop a robust defense framework against privacy attacks.

To address the above concerns, we aim to develop a defense framework that protects DMs from these privacy attacks. **Our key motivation is that fine-tuning DMs on a manipulated self-synthesized dataset could affect their privacy risks.** Despite the finding that fine-tuning DMs on clean self-

synthesized images could increase the privacy risks (Li et al., 2024), it is still under-explored how fine-tuning DMs on a manipulated self-synthesized dataset will affect their privacy risks. We hypothesize that it is possible to construct a manipulated self-synthesized dataset such that fine-tuning on it will mitigate the generative privacy risk of DMs. Therefore, in this work, we aim to take a further step by asking the following research questions:

- **RQ1:** How does fine-tuning DMs on a manipulated self-sythesized dataset affect their generative privacy risks?
- **RQ2:** How to utilize the observations in **RQ1** to develop a defense framework that protects DMs from these privacy attacks?

To answer **RQ1**, we examine the generative privacy risks of DMs under two settings: (1) fine-tuning DM on a clean self-synthesized dataset and (2) fine-tuning on a fully perturbed self-synthesized dataset. We observe that when DMs are fine-tuned solely on self-synthesized clean images, they become more vulnerable to privacy attacks, which is in line with the conclusion in S2L (Li et al., 2024). In contrast, when DMs are fine-tuned on perturbed self-synthesized images become more robust against such attacks but exhibit degraded image generation quality. Motivated by these observations, we answer **RQ2** and propose a simple and effective framework designed to mitigate privacy risks by fine-tuning DMs on a mixed self-synthesized dataset. This dataset is constructed by first generating clean images using domain-specific prompts from the DM, followed by the introduction of adversarial noise to a subset of the images. Furthermore, we develop two mixing strategies: Mixup-I and Mixup-P, which introduce perturbations to images at image-level and pixel-level, respectively. Extensive experimental results demonstrate that our method significantly reduces the efficacy of privacy attacks on DMs while preserving the original image generation quality. To sum up, our contributions are as follows:

- We comprehensively examine how does fine-tuning DMs on a manipulated self-sythesized dataset affect their generative privacy risks, and have the following observations: (1) DMs fine-tuned solely on self-synthesized clean images are more vulnerable to privacy attacks (2) DMs fine-tuned on perturbed self-synthesized images become more robust against privacy attacks but exhibit degraded image generation quality.
- Based on the observations, we propose MixSyn, a simple and effective framework designed to mitigate privacy risks by fine-tuning DMs on a mixed self-synthesized dataset.
- Extensive experimental results demonstrate that our method significantly reduces the efficacy of privacy attacks on DMs while preserving the original image generation quality.

## 2 RELATED WORK

**Diffusion Models (DMs).** Ho et al. (2020) proposed a class of probabilistic generative models that generate samples by initially drawing from a Gaussian distribution and iteratively removing noise to approximate the target data distribution. Starting from an initial image $x_0 \sim q(x)$, the forward process sequentially adds noise at each time step $t \in (0, T)$, producing a series of progressively noisier latent variables $x_0, x_1, ..., x_T$. The reverse process involves training the model $\epsilon_\theta(x_t, t, c)$ to estimate the noise present in $x_t$ and recover $x_{t-1}$. The training objective minimizes the L2 loss between the predicted and true noise during the denoising process, which is defined as:

$$\mathcal{L}_{\text{cond}}(\theta, x_0) = \mathbb{E}_{x_0, t, c, \epsilon \sim \mathcal{N}(0,1)} \left[ \|\epsilon - \epsilon_\theta(x_{t+1}, t, c)\|_2^2 \right] \tag{1}$$

**Customization of DMs.** Recent studies (Ruiz et al., 2023; Zhang et al., 2023; Hu et al., 2021) have developed strategies to cutmoized DMs for personal preference. DreamBooth (Ruiz et al., 2023)is a technique for fine-tuning text-to-image models, like Stable Diffusion, to generate personalized images of a specific subject using a small set of example images. It adjusts the model to incorporate the new subject while preserving the model's general capabilities, enabling the generation of novel images of the subject in various contexts guided by new textual prompts. Instead of altering the model's weights, Textual Inversion (Zhang et al., 2023) learns new embeddings for these concepts, allowing the model to generate personalized and novel images by referencing these embeddings in text prompts. Low-Rank Adaptation (LoRA) (Hu et al., 2021) is a technique used to fine-tune large text-to-image models like Stable Diffusion with minimal computational resources. It works

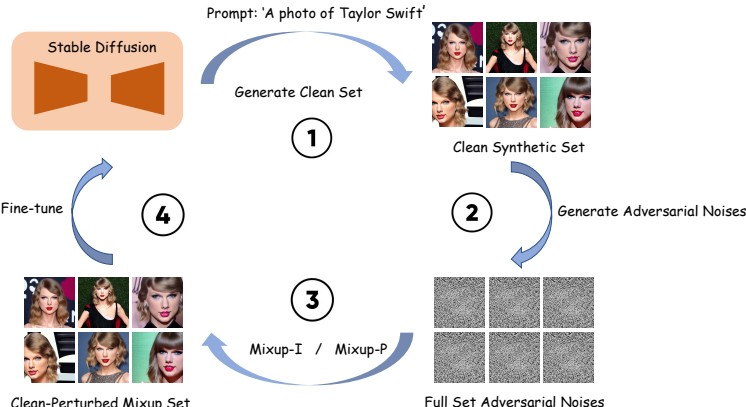

Figure 1: Overview of training pipeline. The first step is to generate a clean synthetic image set with a specific prompt using SD model. The second step is to generate adversarial noises following the procedure as described in Figure 2. The third step is to add the adversarial noises to part of the clean synthetic image set to construct the clean-perturbed mixup set. We propose two ways to add the noise: image-based (Mixup-I) and pixel-based (Mixup-P). Finally, the SD model is fine-tuned on the mixup set to mitigate its generative privacy risks.

by adding low-rank learnable matrices to the model's weights, which are then updated during training. This allows the model to adapt to new concepts or styles with a small number of parameters, preserving the original model's generalization ability while efficiently learning new visual information. It can be integrated with DreamBooth. In our work, we utilize these customization methods to mitigate the generative privacy risks of DMs by fine-tuning them on a specially designed dataset.

**Privacy attacks against generative models.** The privacy risks associated with large generative models have become a significant concern, primarily due to their reliance on vast collections of web images for training, which may inadvertently include sensitive information. Recent studies have demonstrated that diffusion models are particularly susceptible to Membership Inference Attacks (MIAs) (Shokri et al., 2017). In such attacks, an adversary aims to determine whether a given data sample was included in the model's training set (*i.e.*, a member) or as part of the hold-out set (*i.e.*, a non-member). For example, Hu & Pang (2023) employs the loss function $L_{DM}$ to infer the membership status of input samples. Similarly, Wu et al. (2022) explores this vulnerability under the assumption of distinct distributions for member and non-member samples, thereby simplifying the inference task. Furthermore, Carlini et al. (2022) highlights that the privacy risks for diffusion models are substantially greater compared to GAN-based models. More recently, SecMI (Duan et al., 2023) has been introduced as a query-based MIA that assesses membership by analyzing the consistency of posterior estimates during the forward process at each timestep. Additionally, Proximal Initialization Attack (PIA) (Kong et al., 2023) offers a more efficient query-based MIA, leveraging the ground truth trajectory initialized at $t = 0$ and the predicted point to infer membership. These advancements underscore the pressing need for improved privacy protections in diffusion models, as these attacks pose significant concerns to the society.

## 3 HOW DOES FINE-TUNING DMS ON A MANIPULATED SELF-SYNTHESIZED DATASET AFFECT THEIR GENERATIVE PRIVACY RISKS?

In this section, we aim to investigate how could fine-tuning DMs on a manipulated self-synthesized dataset affect their generative privacy risks. First, we introduce the attacking setting for the red team. Next, we introduce the fine-tuning settings. Finally, we examine the generative privacy risks of DMs under two settings: (1) fine-tuning DM on a clean self-synthesized dataset and (2) fine-tuning on a fully perturbed self-synthesized dataset.

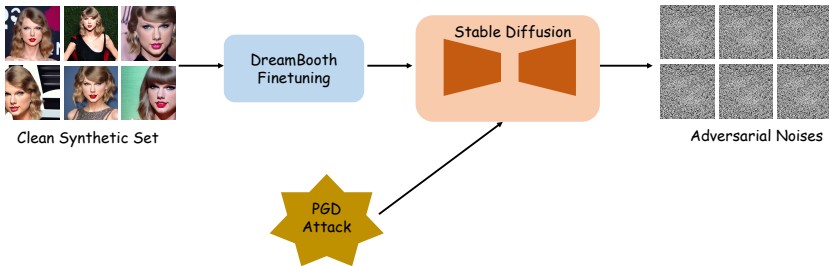

Figure 2: Illustration of the process to generate adversarial noises to the clean synthetic set. We follow a similar procedure as Anti-DreamBooth (Le et al., 2023). We craft the adversarial noise $\delta$ using Projected Gradient Descent (PGD) to maximize the reconstruction loss $\mathcal{L}_{cond}$ of the SD model. The PGD optimization is guided by the process of fine-tuning a fixed SD model on the clean synthetic image set $\mathcal{X}_A$.

## 3.1 ATTACK SETTING

**Threat Model** The threat model assumes the presence of an adversary $A$ who interacts with a pre-trained diffusion model $G$ designed for text-to-image synthesis, with the objective of extracting private information embedded within its training dataset $\mathcal{D}$.

**Victim Model** A conditional diffusion models $G$. The details of the victim model are elaborated in Appendix A.1.

**Adversary Goal** The adversary uses the target prompts $\{p_z\}$ as input, aiming to extract private information linked to the target domains $\mathcal{D}_z$ from the pre-training set $\mathcal{D}$ of $G$. We use Membership Inference Attack (MIA) as the attack objective. Given an image $x^i$, the adversary seeks to determine whether $x^i$ is part of the training set $\mathcal{D}$. The details of the adversary goal is elaborated in Appendix A.1.

**Adversary Capability** We assume that the adversary possesses the capability to manipulate the dataset used for fine-tuning the diffusion model. This assumption is plausible in two scenarios. First, when the diffusion model is publicly accessible, adversaries can execute various operations on the model, including arbitrary fine-tuning. Second, there is a growing trend among model vendors to keep the model parameters confidential while allowing users to upload their own datasets for fine-tuning purposes. For example, OpenAI enables fine-tuning of DALL-E models through an API[1].

**Attack Methods** We use two state-of-the-art Membership Inference Attack (MIA) methods against diffusion models. The detailed introduction of these methods are elaborated in Appendix A.1.

- **Step-wise Error Comparing Membership Inference (SecMI)** (Duan et al., 2023) is a query-based Membership Inference Attack (MIA) that determines membership by evaluating the consistency of forward process posterior estimates at each timestep.
- **Proximal Initialization Attack (PIA)** Kong et al. (2023) is an efficient query-based membership inference attack (MIA) that utilizes the ground truth trajectory obtained by $\epsilon$ initialized at $t = 0$ and the predicted point to infer memberships.

## 3.2 FINE-TUNING SETTING

**Models**: We use the Stable Diffusion v1.4 in our experiment, which is pre-trained on the LAION-Aesthetics subset of LAION-5B. The details can be found in Appendix A.2.

**Datasets**: The SD v1.4 is trained on a subsets of LAION-5B We assume that celebrity pictures represent private domains and investigate whether the SD v1-4 model memorizes these pictures in its pre-training set. As many of the celebrities are also presented in CelebA (Mirjalili et al., 2018; Bortolato et al., 2020; Gupta et al., 2021; Isik & Weissman, 2022), we consider the images in CelebA as the non-private samples. Following the setting in S2L (Li et al., 2024), we construct $40$ private

---

[1]https://platform.openai.com/docs/guides/fine-tuning

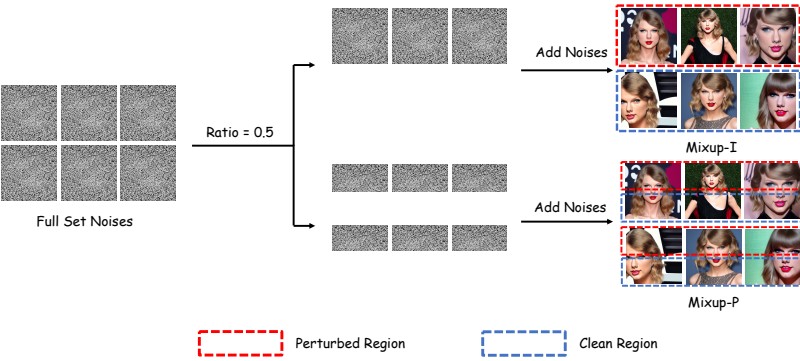

Figure 3: Illustration of two different strategies to add adversarial noises to the clean synthetic set. The strategies are Mixup-I, whose minimum perturbation unit is an image, and Mixup-P, whose minimum perturbation unit is a pixel. In this example, the mixup ratio is set to be 0.5. Therefore, for Mixup-I, the noises will be added to the first three images, whereas for Mixup-P, the noises will be added to half part of every image.

domains corresponding to 40 celebrities with the largest sample sizes in the CelebA dataset. The details can be found in Appendix A.2.

**Evaluation Metrics** Following prior work (Li et al., 2024), we employ AUC and TPR@1%FPR as evaluation metrics for MIA. The higher the AUC and TPR@1%FPR, the more effective the attack is. We utilize the CLIP-R Precision Score (CLIP) as a utility metric to evaluate the alignment between generated images and their corresponding text prompts. We also use BRISQUE (**?**) as image quality metric.

**Finetuning methods** We use three commonly used methods to fine-tune DMs. The detailed introduction of these methods is elaborated in Appendix A.2.

- **Dreambooth** (Ruiz et al., 2023) adjusts the model to incorporate the new subject while preserving the model's general capabilities, enabling the generation of novel images of the subject in various contexts guided by new textual prompts.

- **Textual Inversion** (Zhang et al., 2023) fine-tunes SD by learning new embeddings for these concepts, allowing the model to generate personalized and novel images by referencing these embeddings in text prompts.

- **LoRA** (Hu et al., 2021) is a technique used to fine-tune large text-to-image models like Stable Diffusion with minimal computational resources. It can be integrated with Dream-Booth.

### 3.3 FINETUNING ON CLEAN SELF-SYNTHESIZED IMAGES

In this section, we investigate the first setting where we fine-tune DM on a clean self-synthesized dataset. Follow a similar procedure to prior work (Li et al., 2024), we conduct the following steps:

- **Step 1: Generating Fine-tuning Datasets.** The initial step involves creating a domain-specific fine-tuning dataset by generating a synthetic dataset directly from a pre-trained model $G$ using a target prompt $p_z$ from a private domain $\mathcal{D}_z$.

- **Step 2: Fine-tuning.** We fine-tune the models using standard algorithms on the synthesized set.

- **Step 3: Privacy attacks.** After the model is fine-tuned, we employ MIA (Membership Inference Attack) to attack the model. Given that the adversary focuses on a specific domain, the number of duplicated images in that domain is typically small.

The results in Table 1 show that the pre-trained SD model, without fine-tuning, has the lowest attack success rates for both SecMI (AUC: 0.715, TPR: 0.165) and PIA (AUC: 0.712, TPR: 0.162), indicating lower vulnerability to privacy attacks. Fine-tuning on synthetic data, however, significantly

Table 1: Results on fine-tuning DM on a clean self-synthesized image set. This table shows the effects of perturbation on privacy protection (SecMI and PIA metrics) and image quality (CLIP and BRIS scores).

| Method | SecMI | | | | PIA | | | |
|---|---|---|---|---|---|---|---|---|
| | AUC ↓ | TPR ↓ | CLIP ↑ | BRIS ↑ | AUC ↓ | TPR ↓ | CLIP ↑ | BRIS ↑ |
| Pre-trained | **0.715** | **0.165** | **51.9** | 38.2 | **0.712** | **0.162** | **51.9** | **38.4** |
| DreamBooth | 0.752 | 0.170 | 51.8 | 38.3 | 0.749 | 0.169 | 51.7 | 38.1 |
| Textual Inversion | 0.735 | 0.168 | 51.9 | **38.5** | 0.733 | 0.167 | 51.6 | 38.3 |
| LoRA | 0.741 | 0.167 | 51.7 | 38.0 | 0.738 | 0.166 | **51.9** | 38.2 |
| DreamBooth+LoRA | 0.761 | 0.174 | **51.9** | 38.1 | 0.758 | 0.172 | **51.9** | **38.4** |

Table 2: Results on fine-tuning DM on a perturbed self-synthesized image set. This table shows the effects of perturbation on privacy protection (SecMI and PIA metrics) and image quality (CLIP and BRIS scores).

| Method | SecMI | | | | PIA | | | |
|---|---|---|---|---|---|---|---|---|
| | AUC ↓ | TPR ↓ | CLIP ↑ | BRIS ↑ | AUC ↓ | TPR ↓ | CLIP ↑ | BRIS ↑ |
| Pre-trained | 0.715 | 0.165 | **51.9** | **38.2** | 0.712 | 0.162 | **51.6** | **38.4** |
| DreamBooth | 0.451 | 0.102 | 29.9 | 12.3 | 0.448 | 0.101 | 29.5 | 12.1 |
| Textual Inversion | **0.441** | 0.101 | 31.0 | 11.8 | **0.438** | 0.100 | 30.8 | 11.9 |
| LoRA | 0.445 | **0.100** | 30.1 | 12.0 | 0.442 | **0.099** | 30.4 | 12.2 |
| DreamBooth+LoRA | 0.457 | 0.105 | 30.3 | 12.1 | 0.453 | 0.104 | 30.7 | 12.0 |

increases these risks. This observation is in line with the conclusion in Li et al. (2024). For example, DreamBooth fine-tuning raises SecMI AUC to 0.752 and PIA AUC to 0.749, showing the highest privacy vulnerabilities. Textual Inversion and LoRA also increase risks, with LoRA performing slightly better. Combining DreamBooth and LoRA leads to the highest privacy risks (SecMI AUC: 0.761, PIA AUC: 0.758), highlighting that fine-tuning with synthetic data substantially elevates privacy vulnerabilities.

## 3.4 FINETUNING ON PERTURBED SELF-SYNTHESIZED IMAGES

Next we conduct experiments on perturbed self-synthesized images. Specifically, it takes the following steps:

- **Step 1: Generating Clean Fine-tuning Datasets.** This step is the same as Step 1 in Section 3.3.

- **Step 2: Generating Perturbed Fine-tuning Datasets.** In this step, we add full perturbation masks to the clean fine-tuning datasets. Specifically, we follow a similar procedure as Anti-Dreambooth (Le et al., 2023). The whole process is shown in Figure 2. Denote $\mathcal{P}$ as the clean set of synthetic images generated in the previous step. For each image $p \in \mathcal{P}$, we add an adversarial perturbation $\delta$ to the image $p' = p + \delta$. Then $\mathcal{P}$ is used to finetune a text-to-image generator $\epsilon_\theta$, following the DreamBooth algorithm, to get the optimal hyper-parameters $\theta^*$. The general objective is to optimize the adversarial noise $\Delta_{db} = \{\delta^{(i)}\}_{i=1}^{N_{db}}$ that minimizes the personalized generation ability of that DreamBooth model:

$$\Delta_{db}^* = \arg\min_{\Delta_{db}} \mathcal{A}(\epsilon_{\theta^*}, \mathcal{P}),$$

$$\text{s.t.} \quad \theta^* = \arg\min_{\theta} \sum_{i=1}^{N_{db}} \mathcal{L}_{db}(\theta, p^{(i)} + \delta^{(i)}), \tag{2}$$

$$\text{and} \quad \|\delta^{(i)}\|_p \leq \eta \quad \forall i \in \{1, 2, .., N_{db}\},$$

where $\mathcal{L}_{db}$ is defined in Eq. 6 and $\mathcal{A}(\epsilon_{\theta^*}, \mathcal{X})$ is some personalization evaluation function that assesses the quality of images generated by the DreamBooth model $\epsilon_{\theta^*}$ and the identity correctness based on the synthetic image set $\mathcal{P}$.

The the perturbation $\delta^{*(i)}$ can be calculated with the following objective

$$\delta^{*(i)} = \arg\max_{\delta^{(i)}} \mathcal{L}_{cond}(\theta^*, p^{(i)}), \forall i \in \{1, .., N_{db}\},$$

$$\text{s.t.} \quad \theta^* = \arg\min_{\theta} \sum_{i=1}^{N_{db}} \mathcal{L}_{db}(\theta, p^{(i)} + \delta^{(i)}), \tag{3}$$

$$\text{and} \quad \|\delta^{(i)}\|_p \leq \eta \quad \forall i \in \{1, .., N_{db}\},$$

where $\mathcal{L}_{cond}$ is defined in as Equation 1 and $\mathcal{L}_{db}$ are defined as:

$$\mathcal{L}_{db}(\theta, x_0) = \mathbb{E}_{x_0, t, t'} \|\epsilon - \epsilon_\theta(x_{t+1}, t, c)\|_2^2 + \lambda \|\epsilon' - \epsilon_\theta(x'_{t'+1}, t', c_{pr})\|_2^2 \tag{4}$$

Note that, unlike traditional adversarial attacks, the loss functions are computed only at a randomly chosen timestep in the denoising sequence during training. Then we derive our perturbed synthetic image set $\mathcal{P}'$, such that for each image $p' \in \mathcal{P}'$,

$$p' = p + \delta \tag{5}$$

- **Steps 3: Fine-tuning on perturbed datasets.** This step is the same as Step 2 in Section 3.3. The only difference is that we are fine-tuning on a fully perturbed dataset $\mathcal{P}'$ instead of $\mathcal{P}$.
- **Step 4: Privacy attack** This step is the same as Step 3 in Section 3.3.

Table 2 highlights a trade-off between reducing privacy risks and maintaining image quality, measured by the CLIP metric. Fine-tuning with noise perturbation from Anti-DreamBooth significantly lowers privacy attack success rates compared to models fine-tuned on clean synthetic data. For example, DreamBooth on the perturbed set achieves SecMI AUC of 0.451 and PIA AUC of 0.448, a major improvement from 0.752 and 0.749 in the unperturbed case. Similar reductions in vulnerability are seen with other methods like Textual Inversion and LoRA. However, this reduction in privacy risks comes with a drop in image quality; DreamBooth's CLIP score drops to 29.9 from 49.8, with similar declines for other methods. This shows a clear trade-off: reducing privacy risks via perturbation noise lowers image quality due to introduced artifacts or distortions.

Fine-tuning on synthetic data perturbed by Anti-DreamBooth successfully reduces privacy risks, as demonstrated by lower SecMI and PIA scores. However, this comes at the cost of a significant drop in image quality, as shown by the lower CLIP scores. The results underscore the need for privacy-preserving methods that balance privacy protection with maintaining high-quality outputs in generative models, as will be discussed in the next section.

## 4 METHOD

The motivation behind our proposed framework, **MixSyn**, is to mitigate the increased vulnerability to Membership Inference Attacks (MIA) that arise when diffusion models (DMs) are fine-tuned on clean, self-synthesized datasets. From our earlier observations in Table 1 and Table 2, DMs fine-tuned on clean datasets exhibit higher risks to MIAs, while fine-tuning on fully perturbed datasets significantly reduces privacy risks but at the cost of image quality. Our approach aims to strike a balance by mixing clean and perturbed datasets, preserving image quality while mitigating privacy vulnerabilities.

### 4.1 MIXUP-I AND MIXUP-P

We propose two novel methods, **Mixup-I** and **Mixup-P**, to combine clean and perturbed images in a way that minimizes privacy risks while maintaining the generative quality of DMs:

- **Mixup-I (Image-based Mixing):** This method applies perturbations at the image level. In this approach, a fixed ratio of images in the dataset is perturbed, while the remaining images are left unaltered. This introduces randomness in terms of which images are perturbed, making it harder for MIAs to infer training data while preserving the quality of the non-perturbed images. For instance, with a mixup ratio of 0.5, half of the images in the dataset will be perturbed, reducing overall privacy risks without heavily distorting the generated image set.

Table 3: Performance of Mixup-I at ratio = 0.5. Lower AUC and TPR values represent better privacy protection, while higher CLIP and BRIS scores indicate better image quality.

| Method | SecMI | | | | PIA | | | |
|---|---|---|---|---|---|---|---|---|
| | AUC ↓ | TPR ↓ | CLIP ↑ | BRIS ↑ | AUC ↓ | TPR ↓ | CLIP ↑ | BRIS ↑ |
| Pre-trained | 0.715 | 0.165 | **51.9** | **38.2** | 0.712 | 0.162 | 51.6 | **38.4** |
| DreamBooth | 0.552 | 0.142 | 51.7 | 38.1 | 0.549 | 0.140 | 51.4 | 38.2 |
| Textual Inversion | **0.551** | 0.141 | **51.9** | 38.0 | **0.548** | 0.140 | 51.5 | **38.4** |
| LoRA | 0.550 | **0.140** | 51.6 | 38.0 | 0.547 | **0.139** | **51.7** | 38.2 |
| DreamBooth+LoRA | 0.553 | 0.143 | 51.7 | 38.1 | 0.550 | 0.142 | 51.6 | 38.2 |

- **Mixup-P (Pixel-based Mixing):** Unlike Mixup-I, Mixup-P introduces perturbations at the pixel level within each image. This strategy ensures that no image is fully clean, making it even more challenging for MIAs to differentiate between member and non-member samples. However, because each image contains both clean and perturbed regions, the overall image quality may degrade more noticeably compared to Mixup-I, especially at higher mixup ratios. This method is particularly effective when the goal is to add finer, less perceptible perturbations, but with a higher level of robustness against privacy attacks.

Figure 3 illustrates the differences between these two approaches. The overall training pipeline is shown in Figure 1. By leveraging these two strategies, we can tune the trade-off between privacy and image quality based on the application requirements.

## 5 EXPERIMENT

In this section, we conduct extensive experiments to evaluate the effectiveness of the proposed MixSyn framework in mitigating privacy risks while maintaining high image quality. Specifically, we focus on the performance of the two strategies, **Mixup-I** and **Mixup-P**, across various mixup ratios. We also investigate which components of the model need fine-tuning to achieve optimal trade-offs between privacy protection and image quality.

### 5.1 MIXUP-I

The first set of experiments evaluates Mixup-I, where complete images are perturbed. This approach explores how different mixup ratios affect the trade-off between privacy risks and image quality.

As shown in Figure 4a and Table 3, increasing the mixup ratio reduces the vulnerability of diffusion models (DMs) to Membership Inference Attacks (MIAs). At a 0.5 mixup ratio, the AUC score for SecMI drops from 0.715 (pre-trained) to 0.552 (DreamBooth), highlighting Mixup-I's effectiveness in reducing privacy risks.

Despite the perturbations, image quality remains high, with the CLIP score for DreamBooth at 51.7, close to the pre-trained model's 51.9. This demonstrates that Mixup-I maintains strong image generation quality while reducing privacy risks.

The experiments show a trade-off: lower mixup ratios result in better image quality but higher privacy risks, while higher ratios enhance privacy protection at the cost of image quality. A mixup ratio of 0.5 offers an optimal balance, significantly reducing the AUC for SecMI while preserving image quality close to the pre-trained model.

### 5.2 MIXUP-P

In Mixup-P, pixel-level perturbations ensure that no image is entirely clean, introducing finer granularity compared to Mixup-I. This method aims to obscure membership information while maintaining image quality across various mixup ratios.

As shown in Table 4, Mixup-P offers stronger privacy protection, especially at higher ratios. At a 0.3 mixup ratio, the AUC score for SecMI drops to 0.498, significantly reducing privacy risks compared to the pre-trained model (AUC of 0.715). The finer, distributed noise makes it harder for Membership Inference Attacks (MIAs) to exploit membership information.

Table 4: Performance of Mixup-P at ratio = 0.3. Lower AUC and TPR scores indicate stronger defenses, and CLIP and BRIS scores for image quality, where higher scores represent better alignment and perceptual quality.

| Method | SecMI | | | | PIA | | | |
|---|---|---|---|---|---|---|---|---|
| | AUC ↓ | TPR ↓ | CLIP ↑ | BRIS ↑ | AUC ↓ | TPR ↓ | CLIP ↑ | BRIS ↑ |
| Pre-trained | 0.715 | 0.165 | **51.9** | **38.2** | 0.712 | 0.162 | 51.6 | **38.4** |
| DreamBooth | 0.498 | 0.153 | 51.8 | 38.1 | 0.523 | 0.149 | 51.5 | 38.3 |
| Textual Inversion | **0.465** | 0.132 | **51.9** | 38.0 | **0.482** | 0.128 | **51.7** | **38.4** |
| LoRA | 0.489 | **0.136** | 51.7 | 38.2 | 0.499 | **0.144** | 51.6 | 38.3 |
| DreamBooth+LoRA | 0.506 | 0.145 | 51.6 | 38.1 | 0.517 | 0.141 | 51.8 | 38.2 |

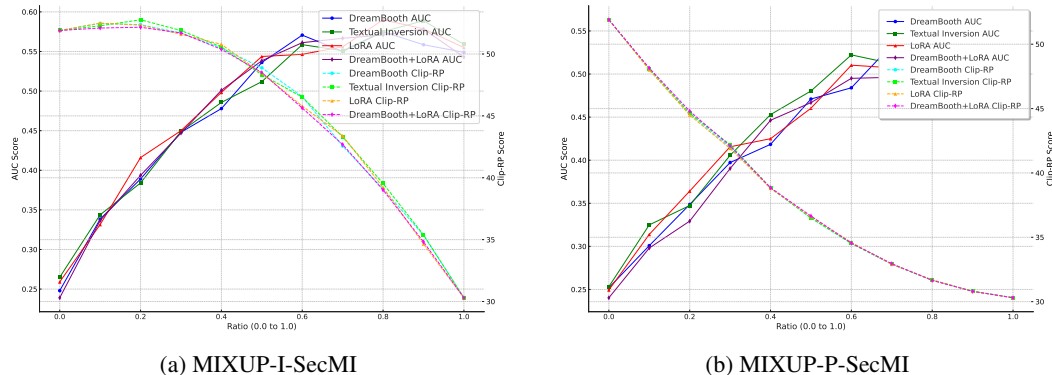

(a) MIXUP-I-SecMI           (b) MIXUP-P-SecMI

Figure 4: Comparison of MIXUP-I and MIXUP-P for SecMI.

However, this comes at a cost to image quality, with a slight decline in CLIP and BRIS scores. For example, DreamBooth's CLIP score at a 0.3 ratio is 51.8, slightly lower than Mixup-I. Pixel-level perturbations introduce subtle artifacts that affect the overall image appearance.

Despite this minor decline, Mixup-P offers stronger privacy protection, making it suitable for scenarios prioritizing security. It strikes a balance, providing enhanced privacy at lower mixup ratios like 0.3 while maintaining acceptable image quality.

### 5.3 WHICH PARAMETERS NEED TO BE FINE-TUNED?

To further optimize the trade-off between privacy protection and image quality, we conducted experiments to determine which specific components of the diffusion model should be fine-tuned. We focused on three key components: the Denoising Network, the Image Encoder, and the Embedding, and evaluated their impact on privacy risks and image quality. Experiment results show that the Denoising Network stands out as the most effective component to fine-tune for privacy protection while fine-tuning the Embedding provides a well-balanced solution. Depending on the specific needs of the application, choosing the right component to fine-tune can optimize the trade-off between privacy protection and image generation quality. All the detailed results and analysis are provided in Appendix A.3

## 6 CONCLUSION

In this paper, We empirically examine the effect of self-synthesized data fine-tuning on DMs regarding their privacy risks. We observe that DMs fine-tuned solely on self-synthesized clean images are more vulnerable to privacy attacks, whereas DMs fine-tuned on perturbed self-synthesized images become more robust against such attacks but exhibit degraded image generation quality. Based on the observation, we propose MixSyn, a simple and effective framework designed to mitigate privacy risks by fine-tuning DMs on a mixed self-synthesized dataset. Extensive experimental results demonstrate that our method significantly reduces the efficacy of privacy attacks on DMs while preserving the original image generation quality. We believe our work takes a further step to the privacy protection of DMs.

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

# A  APPENDIX

## A.1  MORE DETAILED ATTACK SETTING

**Threat Model** The threat model assumes the presence of an adversary $A$ who interacts with a pre-trained diffusion model $G$ designed for text-to-image synthesis, with the objective of extracting private information embedded within its training dataset $\mathcal{D}$.

**Victim Model** Conditional diffusion models $G$ for text-to-image synthesis are gaining increasing popularity due to the accessibility of semantic text inputs, which allow individuals without specialized expertise to generate complex visual content easily. The details of the victim model is elaborated in Appendix.

**Adversary Goals.** The adversary uses the target prompts $\{p_z\}$ as input, aiming to extract private information linked to the target domains $\mathcal{D}_z$ from the pre-training set $\mathcal{D}$ of $G$. We consider two primary attack objectives in the privacy literature: 182 *Membership Inference*: Given an image $x^i$, the adversary seeks to determine whether $x^i$ is part of the training set $\mathcal{D}$. Membership leakage can theoretically correspond to generic privacy leakage under the framework of Differential Privacy Zanella-Béguelin et al. (2023). In certain scenarios, Membership Inference Attacks (MIA) can directly result in a privacy breach. For example, a patient's clinical record could be used to train a disease-associated model. 183 *Data Extraction*: The adversary aims to retrieve training images from $G$ within a targeted domain $\mathcal{D}_z$ associated with a prompt $p_z$.

**Adversary Capabilities.** We assume the attacker has the ability to manipulate the dataset used for fine-tuning the diffusion model. This assumption can hold in two situations: First, if the diffusion model is publicly available, attackers can perform any operations on the model, including arbitrary fine-tuning. Second, there is a growing trend where many model vendors keep the model parameters confidential but permit users to upload data for fine-tuning. For instance, OpenAI allows fine-tuning of DALL-E models via an API[2].

**Attack Methods** We use two state-of-the-art Membership Inference Attack (MIA) methods against diffusion models.

---

[2]https://platform.openai.com/docs/guides/fine-tuning

- **Step-wise Error Comparing Membership Inference (SecMI)** (Duan et al., 2023) is a query-based Membership Inference Attack (MIA) that determines membership by evaluating the consistency of forward process posterior estimates at each timestep.
- **Proximal Initialization Attack (PIA)** Kong et al. (2023) is an efficient query-based membership inference attack (MIA) that utilizes the ground truth trajectory obtained by $\epsilon$ initialized at $t = 0$ and the predicted point to infer memberships.

## A.2 FINE-TUNING SETTING

**Models**: We use the Stable Diffusion v1.4. SD v1.4 is pre-trained on a large-scale dataset, specifically the LAION-Aesthetics subset of LAION-5B, which contains billions of image-text pairs scraped from the internet. This extensive dataset provides a diverse range of semantic content, enabling the model to generate highly detailed and varied images from textual prompts.

**Datasets**: The SD v1.4 is trained on a subsets of LAION-5B We assume that celebrity pictures represent private domains and investigate whether the SD v1-4 model memorizes these pictures in its pre-training set. As many of the celebrities are also presented in CelebA, we consider the images in CelebA as the non-private samples. Following the setting in S2L, we construct $40$ private domains corresponding to 40 celebrities with the largest sample sizes in the CelebA dataset. We define the private domain specified by a domain-specific substring $c_z$ as "¡Celebrity Name¿", and the prompt $p_z$ associated with each private domain $D_z$ is specified as "The face of ¡Celebrity Name¿" with 0.7 possibilities or "A photo of ¡Celebrity Name¿" with 0.3 possibilities.

**Evaluation metrics** In line with S2L, we employ AUC and TPR@1%FPR as evaluation metrics for MIA. For data extraction, we use the count of samples identified as $(10, l_2, 0.1)$-*Eidetic memorization* within the target domain as the evaluation criterion. We also assess the true positive numbers extracted and the precision values averaged across the private domains. Additionally, we utilize the CLIP-R Precision Score (CLIP) as a utility metric to to evaluate the alignment between generated images and their corresponding text prompts. We also use BRISQUE as image quality metric.

**Finetuning methods**

- **Dreambooth** (Ruiz et al., 2023) is a technique for fine-tuning text-to-image models, like Stable Diffusion, to generate personalized images of a specific subject using a small set of example images. It adjusts the model to incorporate the new subject while preserving the model's general capabilities, enabling the generation of novel images of the subject in various contexts guided by new textual prompts. The training loss combines two objectives:

$$\mathcal{L}_{db}(\theta, x_0) = \mathbb{E}_{x_0, t, t'} \|\epsilon - \epsilon_\theta(x_{t+1}, t, c)\|_2^2 + \lambda \|\epsilon' - \epsilon_\theta(x'_{t'+1}, t', c_{pr})\|_2^2 \qquad (6)$$

  where $\epsilon, \epsilon'$ are both sampled from $\mathcal{N}(0, \mathbf{I})$, $x'_{t'+1}$ is noisy variable of class example $x'$ which is generated from original stable diffusion $\theta_{ori}$ with prior prompt $c_{pr}$, and $\lambda$ emphasizes the importance of the prior term.
- **Textual Inversion** (Zhang et al., 2023) is a technique that fine-tunes pre-trained text-to-image models, such as Stable Diffusion, by embedding new concepts into the model using a few example images. Instead of altering the model's weights, Textual Inversion learns new embeddings for these concepts, allowing the model to generate personalized and novel images by referencing these embeddings in text prompts. This approach provides a flexible way to integrate new visual concepts without extensive retraining.
- **LoRA** (Hu et al., 2021) is a technique used to fine-tune large text-to-image models like Stable Diffusion with minimal computational resources. It works by adding low-rank learnable matrices to the model's weights, which are then updated during training. This allows the model to adapt to new concepts or styles with a small number of parameters, preserving the original model's generalization ability while efficiently learning new visual information. It can be integrated with DreamBooth.

## A.3 DETAILS OF THE ABLATION STUDY.

As presented in Table 5, fine-tuning the Denoising Network results in a significant reduction in privacy risks. The AUC for SecMI drops to 0.550, which is on par with fully fine-tuned models like

Table 5: Which parameter to be finetuned? MIXUP-I at ratio=0.3

| Method | SecMI | | | | PIA | | | |
|---|---|---|---|---|---|---|---|---|
| | AUC ↓ | TPR ↓ | CLIP ↑ | BRIS ↑ | AUC ↓ | TPR ↓ | CLIP ↑ | BRIS ↑ |
| Pre-trained | 0.715 | 0.165 | 51.9 | 38.2 | 0.712 | 0.162 | 51.6 | 38.4 |
| DreamBooth | 0.552 | 0.142 | 51.7 | 38.1 | 0.549 | 0.140 | 51.4 | 38.2 |
| Denoising Network | 0.550 | 0.140 | 51.6 | 38.0 | 0.547 | 0.139 | 51.7 | 38.2 |
| Image Encoder | 0.553 | 0.143 | 51.7 | 38.1 | 0.550 | 0.142 | 51.6 | 38.2 |
| Embedding | 0.551 | 0.141 | 51.9 | 38.0 | 0.548 | 0.140 | 51.5 | 38.4 |

DreamBooth. The Denoising Network also maintains relatively high image quality, with a CLIP score of 51.6 and a BRIS score of 38.0. This suggests that the Denoising Network plays a crucial role in mitigating privacy risks while preserving the generative capabilities of the model. Given its ability to reduce privacy risks with minimal impact on image quality, fine-tuning the Denoising Network offers a computationally efficient way to achieve robust privacy protection.

Fine-tuning the Image Encoder provides a slightly different trade-off. While the privacy risk reduction is less pronounced (AUC of 0.553 for SecMI), the Image Encoder preserves higher image quality, particularly in terms of the CLIP score, which remains at 51.7. This makes fine-tuning the Image Encoder a viable option for applications where image quality is a higher priority than privacy protection.

Finally, fine-tuning the Embedding offers a balanced approach, with an AUC of 0.551 for SecMI and a CLIP score of 51.9. This strategy is similar to Textual Inversion, which fine-tunes embeddings to incorporate new concepts into the model. The results indicate that Embedding fine-tuning strikes a good balance between privacy protection and image quality, making it a flexible option depending on the application's requirements.

