# OpenReview forum: "Mitigating Generative Privacy Risks of Diffusion Models via Mixed Self-Synthesized Data Fine-tuning"
_ICLR.cc/2025/Conference — ICLR 2025 Conference Withdrawn Submission_

### Official Review · Reviewer_JyLN · 2024-10-28

**Soundness:** 2
**Presentation:** 2
**Contribution:** 2
**Rating:** 3
**Confidence:** 5

**Summary:**

This paper focuses on the privacy issues about customized diffusion models. Previous research has shown that utilize either natural or synthesized private images can lead to severe privacy issues, These finetuned diffusion models are vulnerable to membership inference attacks. This paper takes a further step to explore the privacy issues in diffusion models finetuned with adversarially watermarked private images. It concludes that these watermarks can not only effectively hinder the customization but also preserve the privacy. Considering the utility-privacy trade-off, this paper mixes the synthesized clean image and watermarked image, hoping to achieve a balance between the privacy and overall image quality.

**Strengths:**

1. This paper is easy to follow and the storyline is quite clear.
2. The paper proposes a method for finetuned diffusion models to defend against MIA, which is very practical in real-world scenarios.

**Weaknesses:**

1. Many experimental details are missing.
* What is the member/hold-out set used in the membership inference attack? Why use the Eidetic memorization samples?
* The details of the image sets leveraged for customization: how many images are there for customization. What does the "largest sample sizes (Line 235)" mean? Why choose largest sample sizes for evaluation?
* Do both the clean image and its corrupted version exist in the training set **or** the clean image and corrupted noise images share different layouts?
* What is the CLIP-R score? Is it the cosine similarity between the text and visual embedding encoded by CLIP?
2. For MIA method, besides SecMI and PIA, the naive attack which leverages the training loss of the diffusion models should also be considered, as the training loss is the most intuitive metric to evaluate the overfitting level.
3. The idea to use a mixed datasets is quite intuitive. However, why not treat these samples differently? For example, set different training objective for noised and clean images. Current design is somewhat simple and seems not optimal.
4. Some experimental results are abnormal.
* In Table 2&3, the AUC is under 0.5 while the TPR is above 0.1, which is quite strange. The ROC curve and log-scaled ROC curve should be provided for further validation.
* What does the AUC of the pretrained model in Table 1&2 mean? The MIA results across the whole datasets or the private datasets?
5. The experiment is limited with SD1.4. Besides, some critical ablations for MIA is missing, for example, the dataset size, the training iters.
6. Some minor typos:
* The reference of BRISQUE is missing.
* Equation 1: $x_{t+1} \rightarrow x_t$

**Questions:**

See weaknesses.

---

### Official Review · Reviewer_VKSN · 2024-11-02

**Soundness:** 3
**Presentation:** 3
**Contribution:** 3
**Rating:** 6
**Confidence:** 3

**Summary:**

The authors propose a method for adversarial attacks on self-generated images from DMs, which is then used to fine-tune DMs to improve their robustness against membership inference attacks.

**Strengths:**

1. Mixup-I and Mixup-P are simple yet effective.
2. The structure and writing of the paper are clear and easy to understand.
3. The paper compares three commonly used DM fine-tuning methods.

**Weaknesses:**

1. Some recent works are overlooked, such as Extracting training data from diffusion models (USENIX Security 2023).
2. There is a lack of mathematical derivation and in-depth theoretical analysis on why Mixup-I and Mixup-P are effective and what differentiates the two.
3. What effect will have if we combine Mixup-I and Mixup-P?
4. There seems to be some randomness in Mixup-I (which samples it is applied to) and Mixup-P (which regions of the image, e.g., top/bottom/left/right). I would suggest conducting more extensive experiments to address this.
5. Validation on only SD v1.4 is insufficient.

**Questions:**

Please refer to Weakness.

---

### Official Review · Reviewer_chUC · 2024-11-02

**Soundness:** 1
**Presentation:** 1
**Contribution:** 2
**Rating:** 3
**Confidence:** 3

**Summary:**

The paper proposes a method to reduce the privacy risks of diffusion models against membership inference attacks (MIAs). By fine-tuning the model on self-synthesized data with adversarial perturbations and combining it with clean self-synthesized data using Mixup, the diffusion model achieves improved resistance to MIA.

Experiments demonstrate that the proposed method, using PGD adversarial perturbations, enhances the performance of the Stable Diffusion 1.4 model on the CelebA dataset.

**Strengths:**

1. The paper introduces a novel method - perturbed self-synthesized fine-tuning with mixup - to improve diffusion models' resistance to MIA while preserving image quality. The mixup method compensates for any potential drop in quality, which is a notable contribution.
2. The experiments comprehensively consider various fine-tuning methods (Textual Inversion, LoRA, DreamBooth, and DreamBooth + LoRA) to address RQ1. The results validate the hypothesis that self-synthesized fine-tuning increases vulnerability to attacks, with DreamBooth + LoRA proving the most effective.
3. The paper tests the proposed method against two MIAs, demonstrating its effectiveness at certain mixup ratios. Figure 4 provides valuable insights into the tradeoff between AUC and Clip Score across different ratios.

**Weaknesses:**

1. The experimental design lacks comprehensiveness, especially in terms of (1) perturbation settings, (2) fine-tuning datasets and MIA evaluation (only CelebA is used), and (3) diffusion model variety. The core of the method are the manipulation techniques; however, only one perturbation method (PGD noise added to the fine-tuning data) is examined.
2. Additionally, proposed mixup perturbation is limited in using self-synthetic data. For instance, does RQ1 also hold for real datasets? Are the findings in Tables 1 and 2 applicable to real datasets? Does the approach assume that real data cannot be used during defense?
3. While the paper aims to mitigate privacy risks in diffusion models, it does not clearly define the privacy attack victims. Two potential victims could be (1) the developer of the diffusion model, who may face privacy risks or legal issues if an MIA is successful, and (2) users who fine-tune the model with private images, as a successful MIA could expose their inputs.
4. The positioning of privacy benefits is unclear. The mention of artists forming unions to combat unauthorized use of their work suggests an ethical angle, implying that this defense could prevent artists from verifying or contesting such use in court, which may be inappropriate in this context.
6. The privacy risk is evaluated through MIA, which has a limited scope (e.g., reducing privacy risks for specific target-domain samples such as 40 celebrities). This point is included in the appendix but should be moved to the main text with justification on the exact privacy improvement.
7. The paper lacks an in-depth exploration of why mixing is effective.
8. Figure/Table placement is not good. For instance, Figure 2 is on the top of page 4 but first mentioned on page 6, and Figure 3 on page 5 is first mentioned on page 8. Tables 3 and 4 should also be placed closer together.
9. Writing clarity could be improved in several areas:
    - The caption for Table 1, "This table shows the effects of perturbation on privacy protection," is inaccurate as no perturbations are applied in this table.
    - Line 307: "we add full perturbation masks to the clean fine-tuning datasets"—why is additive noise referred to as "masks"?
    - Line 311: "following the DreamBooth algorithm"—DreamBooth should be presented as an example rather than the default, as Table 2 suggests that the method is agnostic to the fine-tuning approach.
    - Figure 4 could benefit from a more descriptive caption.

**Questions:**

Can the author respond to the weaknesses that I raised?

**Details Of Ethics Concerns:**

The mention of "artists increasingly form unions to combat the unauthorized use of their works by commercial generative models, the need for thorough security assessments and risk evaluations prior to the public release of these models becomes ever more pressing." in the context of company develop DM models that uses artists' original work, implies that the use of this proposed defense could prevent artists from verifying or contesting such use in court, which may be inappropriate in this context and considered as harmful thoughts.

There could be much better example to show the importance of mitigating privacy risks.

---

### Official Review · Reviewer_BNNM · 2024-11-03

**Soundness:** 3
**Presentation:** 3
**Contribution:** 2
**Rating:** 5
**Confidence:** 3

**Summary:**

This paper presents MixSyn, a framework for mitigating privacy risks in diffusion models through fine-tuning on mixed self-synthesized datasets. The key contribution is a mixing strategy that combines clean and perturbed synthetic images at both image and pixel levels to balance privacy protection and generation quality. The authors demonstrate empirical improvements in privacy protection while maintaining image quality through extensive experiments.

**Strengths:**

- The paper presents a systematic investigation of privacy risks in diffusion models:
  - Demonstrates that fine-tuning on clean synthetic data increases privacy vulnerability (SecMI AUC from 0.715 to 0.752)
  - Shows that fully perturbed data improves privacy but degrades quality (CLIP score drops from 51.9 to 29.9)
  - These findings highlight the importance of a balanced approach

- The two-level mixing strategy has the benefits:
  - Image-level mixing (Mixup-I) offers general control while maintaining high image quality
  - Pixel-level mixing (Mixup-P) provides more detailed privacy protection
  - The combination allows flexible trade-offs between privacy and quality:
    - Mixup-I achieves 0.552 SecMI AUC while maintaining 51.7 CLIP score
    - Mixup-P reaches 0.498 SecMI AUC with 51.8 CLIP score

**Weaknesses:**

- The paper claims that maximizing $L_{cond}$ through PGD can protect privacy, but lacks theoretical justification for why this specific loss function helps prevent membership inference:
  - The connection between noise patterns generated by PGD and privacy protection needs formal analysis
  - No definite limits are provided on the privacy guarantees under this noise generation scheme

- The partial perturbation approach raises a few theoretical questions:
  - For Mixup-I, leaving some images completely unperturbed could create privacy risk, as these images might still leak training information
  - The paper doesn't analyze how information from perturbed images might indirectly leak through the unperturbed portions of the dataset

- The proposed method, while practical, doesn't introduce any new technical ideas:
  - The noise generation method directly uses PGD from Anti-DreamBooth without major improvements
  - The mixing strategies, while effective, employ straightforward random selection without advanced methods
  - No new design ideas or improvement methods are introduced to the fine-tuning process

**Questions:**

- 1. Given that $L_{cond}$ is designed for image reconstruction, what is your theoretical justification for using it as a privacy protection metric?
  - How does maximizing this loss specifically prevent membership inference?

- 2. Your method allows some images/regions to remain unperturbed. How do you ensure this doesn't create privacy vulnerabilities?
  - What is the minimum ratio of perturbation needed to guarantee privacy protection?
  - Have you analyzed how information might leak through the interaction between perturbed and unperturbed portions during training?

---

### Official Review · Reviewer_XQVg · 2024-11-04

**Soundness:** 3
**Presentation:** 3
**Contribution:** 2
**Rating:** 3
**Confidence:** 4

**Summary:**

This paper focuses on the trade-off between increasing privacy risks and the degradation of generation quality when fine-tuning text-image diffusion models. It demonstrates that while tuning a diffusion model on perturbed data can mitigate privacy leaks, it adversely affects the generation quality of the model.
To address this, the paper introduces MixSyn, which utilizes a blend of clean and perturbed data for model fine-tuning. This approach aims to balance the preservation of privacy and the maintenance of generation quality. Experimental results validate that MixSyn effectively protects against privacy leaks in the context of two specific membership inference attacks while sustaining the quality of the generated images.

**Strengths:**

1. The paper is easy to read.
2. The motivation behind MixSyn is clearly articulated, and the method is straightforward.
3. The experiments effectively demonstrate that MixSyn can simultaneously mitigate privacy risks and maintain generation quality of the fine-tuned models.

**Weaknesses:**

1. The task setting lacks practical value. S2L [1] reveals that fine-tuning the pre-trained models with manipulated data can amplify the existing privacy risks. In this context, fine-tuning is merely a component of a attack strategy targeting pre-trained models. As for the quality of the generated content from the fine-tuned mode, it doesn't matter. What matters is the ability to further extract information from the model. While the tuned model demonstrates reduced privacy risks and improved generative capabilities, vulnerabilities in the pre-trained model persist. Consequently, an adversary could still exploit these vulnerabilities using methods like S2L to extract private information from the pre-trained models.
2. The novelty is limited. The primary innovation reported is that a diffusion model fine-tuned on a perturbed self-synthetic dataset presents reduced privacy risks. However, the strategy used for perturbing the self-synthetic data is derived from existing work, specifically Anti-Dreambooth [2].
3. In continuation of the previous comment, several other methods, such as PhotoGuard [3], AdvDM [4], and Mist [5], offer alternative approaches to adding perturbations to images. These techniques could be integrated into MixSyn, potentially enhancing its effectiveness in perturbing images for fine-tuning diffusion models. A comparative analysis of these perturbation strategies could significantly strengthen the findings and provide a more comprehensive evaluation of MixSyn's capabilities.

[1] Li et al. Shake to Leak: Fine-tuning Diffusion Models Can Amplify the Generative Privacy Risk

[2] Le et al. Anti-dreambooth: Protecting users from personalized text-to-image synthesis.

[3] Salman et al. Raising the Cost of Malicious AI-Powered Image Editing.

[4] Liang et al. Adversarial Example Does Good: Preventing Painting Imitation from Diffusion Models via Adversarial Examples.

[5] Liang et al. Mist: Towards Improved Adversarial Examples for Diffusion Models.

**Questions:**

See in Weaknesses.

---

### Note · Authors · 2024-11-15

**Comment:**

We appreciate the reviewer for their feedbacks.

**Withdrawal Confirmation:**

I have read and agree with the venue's withdrawal policy on behalf of myself and my co-authors.